# ReDeRef: Recursive Delegation and Reflection for Multi-Turn LLM Agent Collaboration with Dynamic Capability Discovery

## Abstract

Large language models (LLMs) achieve strong single-turn performance, yet real-world deployments demand multi-turn, multi-agent coordination with dynamic routing, reliable credit assignment, and long-horizon memory. We introduce ReDeRef, a lightweight, training-free framework that wraps arbitrary LLM agents with four synergistic components: (i) online Bayesian delegation (Thompson sampling) for dynamic routing; (ii) calibrated self-reflection via an LLM judge to drive credit assignment and recursive re-routing; (iii) text-appropriate aggregation using selection with evidence checks; and (iv) memory-aware belief updates for long-term adaptation. Across domain-diverse, split-knowledge tasks, ReDeRef attains higher success rates than static non-collaborative baselines, with ablations indicating that the recursive re-routing loop contributes the majority of gains on initially failed tasks while online Bayesian updates improve routing efficiency. These results suggest that an interpretable, probabilistic wrapper can substantially enhance multi-agent LLM coordination enabling dynamic task routing, emergent specialization, and long-term adaptability with minimal overhead.

## 1 Introduction

Large Language Models (LLMs) such as GPT-4, Gemini, and Claude have evolved from autocomplete curiosities into broadly capable reasoning engines, surpassing human performance on tasks ranging from legal exams to software engineering challenges. However, most of these benchmarks are *single-turn* interactions. Real-world deployments, in contrast, demand persistence and adaptability: evolving software repositories over weeks, synthesizing iterative scientific reviews, or mediating multi-stakeholder corporate decisions. These are inherently multiturn, long-horizon, and collaborative, pushing beyond what any single monolithic model, no matter how large, can reliably achieve.

Multi-agent LLM systems have emerged as a promising direction to scale intelligence by composing the complementary skills of multiple agents. For example, one agent might specialize in writing unit tests, another in literature surveys, and a third in policy analysis. By enabling interaction between such experts, these systems can address tasks outside the scope of any individual model. Yet, their deployment remains hampered by three persistent bottlenecks:

**(1) Dynamic task routing.** Existing orchestrators often rely on fixed pipelines or static vector-similarity rules. These approaches are brittle: When task requirements change or agent performance degrades, the system continues to misroute, invoking wrong experts and compounding errors.

**(2) Credit assignment across long horizons.** In extended dialogues, failures may not be visible until dozens of turns later. Without timely and fine-grained feedback, the system cannot effectively demote underperforming agents or up-weight reliable ones, leading to stagnation in routing efficiency.

**(3) Context fragility.** Naïve concatenation of conversation history inflates the prompt length, increases computational cost, and risks catastrophic forgetting of an important early context.

We propose ReDeRef(Recursive Delegation and Reflection), a lightweight and training-free controller that directly addresses these challenges. ReDeRef wraps any pool of agents with four synergistic components: (1) **Bayesian delegation** using Thompson sampling to maintain and update agent

competence beliefs in real time; (2) **calibrated self-reflection** via an LLM judge that drives credit assignment and recursive re-routing; (3) **evidence-checked aggregation** that ensures responses are selected, not merely averaged; and (4) **memory-aware priors** that mitigate cold-start issues and prevent context bloat.

By combining these elements into a recursive loop, REDEREF transforms a set of independent agents into an adaptive collective. Experiments across split-knowledge tasks show that this approach yields emergent specialization, synergy, and adaptability with minimal overhead.

**Contributions.** We present REDEREF, a training-free, probabilistic controller for multi-agent LLM collaboration that: (i) performs Bayesian delegation via Thompson sampling for dynamic, uncertainty-aware routing; (ii) integrates an LLM/programmatic judge whose calibrated binary verdicts drive both credit assignment and recursive re-routing; (iii) aggregates by *selection with evidence* rather than averaging; and (iv) seeds memory-aware priors for fast cold-start. On public agent benchmarks (WebArena, Mind2Web/GAIA, SWE-bench subsets) and our split- knowledge suite, REDEREFmatches or exceeds strong reflection/ensemble baselines (Reflexion, Self-Refine, ReAct, Tree-of-Thoughts, AutoGen, Mixture-of-Agents) while reducing agent calls and tokens by 10–30% at matched success, and it adapts rapidly under agent or judge impairments.

## 2 RELATED WORK

**Reasoning, reflection, and search.** Methods such as ReAct Yao et al. (2023a), Self-Refine Madaan et al. (2023), Reflexion Shinn et al. (2023), and Tree-of-Thoughts (ToT) Yao et al. (2023b) enhance reasoning by coupling tool use, iterative self-feedback, episodic memory, or backtracking search. Agent-R Park et al. (2023) and recent surveys Wang et al. (2024) highlight reflection as central to multi-agent LLMs. Unlike these, REDEREF uses an *explicit probabilistic controller*: Thompson sampling delegates under uncertainty, while binary judge outcomes update interpretable Beta posteriors that drive re-routing.

**Orchestration, ensembles, and routing.** Frameworks such as AutoGen Wu et al. (2023) coordinate agents via scripted protocols, and Mixture-of-Agents (MoA) Jiang et al. (2024) aggregate outputs via ensembles. Early orchestrators (e.g., LangChain graphs, AutoGPT workflows) hard-code pipelines, while systems like RopMura Liu et al. (2024) or MLPO Chen et al. (2023) add dynamic routing but require central training. In contrast, REDEREF performs online Bayesian delegation without retraining, maintaining efficiency and decentralization.

**Learning-based coordination.** Multi-agent reinforcement learning has long addressed coordination Lowe et al. (2017); Foerster et al. (2018), with adaptations for LLMs such as SWEET-RL Zhang et al. (2023). Yet RL methods are sample-hungry. Probabilistic approaches, e.g., "Too Many Cooks" Kumar et al. (2022), estimate expertise from sparse data. REDEREF extends this line with Thompson sampling, cooldown, and memory-aware priors, offering lightweight, interpretable credit assignment.

**Benchmarks and positioning.** Public environments (WebArena Zhou et al. (2023), Mind2Web Deng et al. (2023), GAIA Parisi et al. (2023), SWE-bench Jimenez et al. (2023)) highlight the difficulty of real-world, long-horizon tasks (e.g., GPT-4 achieves ∼14% on WebArena vs. humans at ∼78%). Against this backdrop, REDEREF contributes a training-free, adaptive controller that combines Bayesian delegation, embedded reflection, and memory to improve efficiency, robustness, and interpretability over static, ensemble, or RL-based approaches.

## 3 THE REDEREF FRAMEWORK

We formalize multi-agent coordination as a recursive Bayesian decision process. Given a task $T$ with query $q$ and a population of heterogeneous agents $N$ $A = \{A_1, \ldots, A_N\}$, each agent $A_i$ possesses an unobserved task-conditional competence $\theta_i \in [0, 1]$. The controller maintains a Beta posterior $\theta_i \sim \text{Beta}(\alpha_i, \beta_i)$, updated online from binary feedback $y \in \{0, 1\}$ indicating success or failure on judged outcomes. Unless otherwise stated, priors are $\alpha_0 = \beta_0 = 1$ (uninformative) or initialized

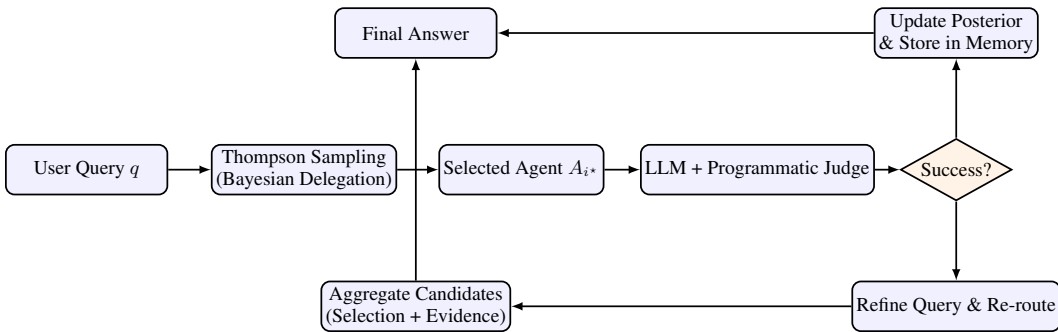

Figure 1: Straight-line flow for REDEREF. Queries pass through delegation, agent execution, and judging. Success flows upward through memory into the final answer, while failure flows downward into refinement and aggregation before reaching the final answer.

by the memory-aware scheme in section 3.6. The overall decision-making pipeline of REDEREF, including Bayesian delegation, recursive reflection, and memory-based adaptation, is illustrated in Figure 1. This flowchart highlights how success cases are propagated through posterior updates and memory, whereas failures trigger refinement, aggregation, and rerouting until a satisfactory solution is produced.

### 3.1 BAYESIAN DELEGATION VIA THOMPSON SAMPLING (CORE POLICY)

At recursion depth $d$ for query $q_d$, REDEREF treats agent selection as a multi-armed bandit problem and applies Thompson sampling:

$$\hat{\theta}_i \sim \text{Beta}(\alpha_i, \beta_i), \qquad i^\star \in \arg\max_i \hat{\theta}_i. \tag{1}$$

The selected agent $A_{i^\star}$ proposes a candidate solution. After judging, the posterior updates in closed form:

$$\alpha_{i^\star} \leftarrow \alpha_{i^\star} + y, \qquad \beta_{i^\star} \leftarrow \beta_{i^\star} + (1 - y), \tag{2}$$

where $y = \mathbb{I}[\text{success}]$. Thompson sampling provides a principled exploration–exploitation trade-off: agents with higher posterior means are preferred while uncertainty induces exploration.

**Variant (for ablations only).** For comparison in ablations, we evaluate a Boltzmann policy over posterior means $\mu_i = \alpha_i/(\alpha_i + \beta_i)$ with temperature $\tau$. This variant is not used in the default system.

### 3.2 SELF-REFLECTION AND JUDGING

Each candidate is evaluated by a judge $J$ that yields a calibrated binary verdict $E \in \{\text{SUCCESS, FAILURE}\}$. Two evidence channels are integrated:

1. **Programmatic metrics**, when available (e.g., EM/F1, supporting-facts F1, unit-test pass rate), which short-circuit to success on unambiguous positives.

2. **LLM adjudication**, which produces a binary decision and a brief rationale when metrics are absent or inconclusive.

We calibrate $J$ on a small labeled set ($N \approx 200$) to estimate FP/FN rates and set thresholds. The verdict is mapped to the Bernoulli feedback driving Bayesian updates:

$$y = \mathbb{I}[E = \text{SUCCESS}].$$

This makes reflection the engine of both *credit assignment* (posterior updates) and *control* (whether to re-route).

### 3.3 TEXT-APPROPRIATE AGGREGATION

REDEREF aggregates by *selection with evidence*, not averaging. We construct a set of candidates with rationales, score them using task metrics or retrieval-grounded entailment checks, optionally run a bounded debate (up to two refinement turns) among top candidates, and select the best-supported answer.

For structured outputs (e.g., numeric fields, dates, JSON), we extract atomic fields, fuse them via competence-weighted voting using $\mu_i = \alpha_i/(\alpha_i + \beta_i)$ as trust weights, then regenerate a coherent output. This modality-aware design yields robustness across open-ended and structured tasks while maintaining interpretability.

### 3.4 RECURSIVE RE-ROUTING

If the judged outcome is FAILURE, REDEREF executes a budgeted recursive step: (i) update $\text{Beta}(\alpha_{i^\star}, \beta_{i^\star})$ with $y = 0$, (ii) refine the query with the judge's critique, and (iii) re-route via Thompson sampling to the next most promising expert. The recursion terminates upon any of the following:

1. **Success**: the judge returns SUCCESS;

2. **Depth limit**: reaching a maximum recursion depth $D$;

3. **Budget exhaustion**: cumulative cost (tokens, time) exceeding $B$;

4. **Plateau**: no judged improvement across recent iterations.

This procedure is not full tree search (no rollout value estimation) but a lightweight, online mechanism that reliably recovers from local errors and discovers productive multi-agent chains with low overhead. The overall recursive procedure of REDEREF, combining Bayesian delegation, self-reflection, query refinement, and memory-aware updating, is summarized in Algorithm 1.

### 3.5 FORMALISM (STEP-BY-STEP)

The step-by-step flow below corresponds directly to the operations detailed in Algorithm 1. **Step 1: Probabilistic Delegation.** At depth $d$, sample $\hat{\theta}_i$ from each agent's Beta posterior and delegate to $A_{i^\star}$ with the largest draw. This induces greedy behavior when one agent is clearly superior and exploration when posteriors are uncertain.

**Step 2: Information Merging.** Each responding agent returns an output $o_j$ with rationale. Candidates are scored using programmatic metrics and/or evidence-grounded entailment. For competitive candidates, a bounded debate permits two rounds of critique and revision. The final response $O_d$ is chosen by selection, not averaging.

**Step 3: LLM-Driven Self-Reflection.** The judge $J$ evaluates $O_d$ with respect to accuracy and completeness, returning $E \in \{\text{SUCCESS}, \text{FAILURE}\}$ and rationale $\rho$. The binary update variable is $y = \mathbb{I}[E = \text{SUCCESS}]$, which directly updates $(\alpha, \beta)$.

**Step 4: Recursive Re-Routing.** Upon failure, REDEREF refines the query using $\rho$, enforces a cooldown $r$ on the last agent to avoid immediate reselection, and re-routes via Thompson sampling subject to budget and depth constraints.

**Step 5: Memory and Belief Update.** Posterior updates accumulate over time, enabling emergent specialization. Memory $\mathcal{M}$ stores tuples $(x, i, y, \rho, t)$ for analysis and future prior seeding.

### 3.6 MEMORY-AWARE PRIORS AND COLD-START MITIGATION

To reduce early-round inefficiency, we seed priors with similarity- and recency-weighted historical outcomes:

$$\alpha_i \leftarrow \alpha_0 + \sum_m K(x_d, x_m)\, y_m\, w_{\Delta t_m}, \qquad \beta_i \leftarrow \beta_0 + \sum_m K(x_d, x_m)\,(1 - y_m)\, w_{\Delta t_m}, \quad (3)$$

---

**Algorithm 1** REDEREF: Recursive Delegation and Reflection (final)

---

**Require:** Query $q$, agents $A_1..A_N$ with $(\alpha_i, \beta_i)$, memory $\mathcal{M}$, judge $J$, max depth $D$, budget $B$, cooldown $r$
1: $x \leftarrow \text{embed}(q)$
2: initialize $(\alpha_i, \beta_i)$ via memory-aware priors from $\mathcal{M}$
3: $\text{cool}[i] \leftarrow 0$ for all $i$;   $\text{spent} \leftarrow 0$
4: $\mathcal{C} \leftarrow \emptyset$                                                              ▷ set of all candidates with evidence
5: $(\text{best}, \text{bestScore}) \leftarrow (\varnothing, -\infty)$
6: **for** $d = 1$ to $D$ **do**
7:     **if** $\text{spent} \geq B$ **then break**
8:     **end if**
9:     **Sampling:** for each $i$, draw $\hat{\theta}_i \sim \text{Beta}(\alpha_i, \beta_i)$ if $\text{cool}[i] = 0$ else set $\hat{\theta}_i \leftarrow -\infty$
10:     **if** $\max_i \hat{\theta}_i = -\infty$ **then**                          ▷ all cooling; force one step of exploration
11:         set $\text{cool}[j] \leftarrow 0$ for $j = \arg\max_i \text{cool}[i]$                ▷ or the smallest cooldown
12:         **continue**
13:     **end if**
14:     $i^\star \leftarrow \arg\max_i \hat{\theta}_i$                                      ▷ tie-break by larger $\alpha_i/(\alpha_i + \beta_i)$
15:     $(\text{cand}, \text{usage}) \leftarrow A_{i^\star}(q)$
16:     $\text{spent} \mathrel{+}= \text{usage}$
17:     **if** $\text{spent} > B$ **then break**
18:     **end if**
19:     $\text{score}_{\text{prog}} \leftarrow \text{programmatic\_metrics}(q, \text{cand})$
20:     **if** $\text{score}_{\text{prog}}$ is unambiguous positive **then**
21:         $E \leftarrow \text{SUCCESS};$    $\rho \leftarrow$ "programmatic pass"
22:     **else**
23:         $(E, \rho, \text{score}_{\text{judge}}) \leftarrow J(q, \text{cand}, \text{score}_{\text{prog}})$
24:     **end if**
25:     $y \leftarrow \mathbb{I}[E = \text{SUCCESS}]$
26:     $\alpha_{i^\star} \mathrel{+}= y;$    $\beta_{i^\star} \mathrel{+}= (1 - y)$
27:     $\mathcal{M} \leftarrow \mathcal{M} \cup \{(x, i^\star, y, \rho, \text{now}())\}$
28:     $\mathcal{C} \leftarrow \mathcal{C} \cup \{(\text{cand}, \text{score}_{\text{prog}}, \text{score}_{\text{judge}}, i^\star)\}$
29:     $\text{candScore} \leftarrow \text{combine\_scores}(\text{score}_{\text{prog}}, \text{score}_{\text{judge}})$
30:     **if** $\text{candScore} > \text{bestScore}$ **then**
31:         $(\text{best}, \text{bestScore}) \leftarrow (\text{cand}, \text{candScore})$
32:     **end if**
33:     **if** $y = 1$ **then**
34:         **return** $\text{aggregate\_select}(\mathcal{C}, \{\mu_i = \alpha_i/(\alpha_i + \beta_i)\})$           ▷ selection with evidence
35:     **end if**
36:     $\text{cool}[i^\star] \leftarrow r;$   **for all** $j$**:** $\text{cool}[j] \leftarrow \max(0, \text{cool}[j] - 1)$
37:     $q \leftarrow \text{refine}(q, \text{cand}, \rho)$
38:     **if** $\text{plateau}(\mathcal{C}, k)$ **then break**                    ▷ no score improvement over last $k$ attempts
39:     **end if**
40: **end for**
41: **if** $\mathcal{C} \neq \emptyset$ **then**
42:     **return** $\text{aggregate\_select}(\mathcal{C}, \{\mu_i\})$                ▷ best-so-far with evidence checks
43: **else**
44:     **return** FAILURE                                              ▷ no candidate produced
45: **end if**

---

where $x_d = \text{embed}(q_d)$, $K$ is a task-similarity kernel (e.g., cosine over sentence embeddings), and $w_{\Delta t_m} = \exp(-\lambda \Delta t_m)$ applies temporal decay. This initialization biases competence posteriors toward agents that recently succeeded on similar tasks while preserving adaptability under drift.

**Remark (Interpretability).**   The Beta parameters $(\alpha_i, \beta_i)$, the judge rationales $\rho$, and the selection history (agents chosen, verdicts, costs) constitute an auditable decision trail, facilitating error analysis and responsible deployment.

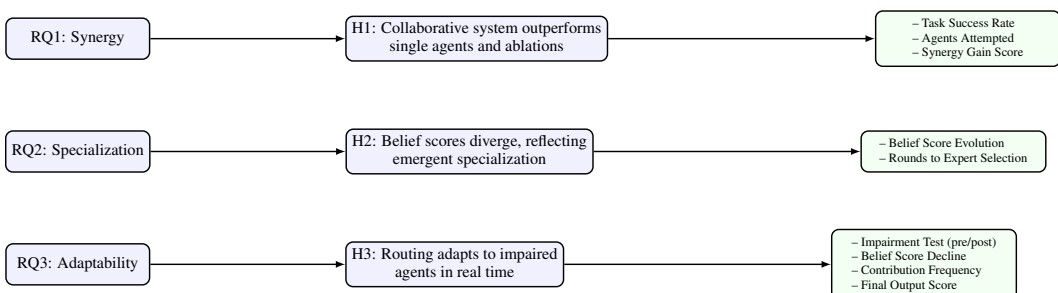

Figure 2: Flow diagram linking research questions (RQ), hypotheses (H), and evaluation metrics, illustrating the conceptual alignment between objectives, testable claims, and operational measures.

### 3.7 SUMMARY

REDEREF operationalizes multi-agent LLM collaboration as a recursive Bayesian controller. Thompson sampling ensures principled routing, reflection yields calibrated credit assignment, selection-based aggregation promotes evidence-grounded robustness, and memory-aware priors provide continuity across tasks. The resulting system is lightweight, interpretable, and capable of emergent specialization, synergy, and adaptation under realistic budget constraints.

## 4 EXPERIMENTAL DESIGN FOR DEMONSTRATING EMERGENCE

Figure 2 provides an overview of our evaluation design, linking research questions to hypotheses and associated metrics. This diagram emphasizes the structured alignment between conceptual goals—*synergy*, *specialization*, and *adaptability*—their testable hypotheses, and the quantitative measures used to validate them.

To assess the effectiveness of REDEREF in fostering emergent collaborative behaviors, we designed a rigorous experimental protocol built around these three dimensions. Each dimension corresponds to a guiding research question, a specific hypothesis, and a set of operational metrics.

**Scope and Appendix.** For brevity, we present only the high-level structure and hypotheses in this section (see Fig. 2). Detailed descriptions of the experimental protocol are provided in the Appendix: (i) the task-generation pipeline (*Stages 1–3*) is described in *Evaluation Tasks*; (ii) the full agent zoo specification is provided in *Agent Population*; (iii) definitions of baselines and ablation protocols are in *Baselines and Ablation Studies*; and (iv) metric formulations and computation procedures for *performance*, *specialization* (H2), *adaptability* (H3), and *synergy* (H1) are outlined in *Metrics for Emergent Behaviors*.

This section therefore focuses on the conceptual mapping from research questions to hypotheses and the primary validation measures.

## 5 RESULTS AND ANALYSIS

We summarize both *effectiveness* (success, quality) and *efficiency/robustness* (tokens, calls, latency, regret, and adaptation). While recursion alone is surprisingly strong on our split-knowledge suite, REDEREF consistently shifts the *efficiency frontier*: at matched success it uses fewer tokens and agent calls, reaches first success faster, and adapts under drift, whereas random delegation wastes capacity on unreliable agents (see budget-matched Pareto curves and regret plots).

### 5.1 OVERALL PERFORMANCE

Table 1 reports output quality across delegation strategies. REDEREFachieves a task success rate of 96.65% (±0.8), a margin of over 16 percentage points compared to the *Single Best Agent (Oracle)* baseline at 80.45% (±1.1). Importantly, this oracle baseline assumes perfect knowledge of agent competence, yet still underperforms because collaboration across multiple experts is essential in our

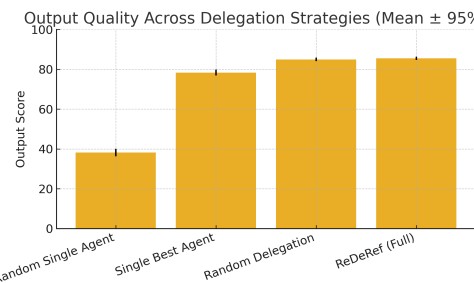

(a) Task success across delegation strategies (mean ± 95% CI).

(b) Output quality across delegation strategies (mean ± 95% CI).

Figure 3: Comparison of delegation strategies using (a) task success and (b) output quality.

split-knowledge tasks. The *Random Single Agent* baseline underperforms substantially at 18.99% (±1.3), confirming that multi-agent coordination is indispensable. The *Random Delegation* ablation performs strongly at 96.46% (±0.7), reflecting the benefit of recursion alone, but remains less efficient than belief-guided routing. Together, these comparisons establish that ReDeRef's probabilistic delegation provides measurable gains over naive recursion.

Table 1: Output quality comparison across delegation strategies. Results are mean ± 95% CI across evaluation tasks.

| Method | Output Score | Task Success (%) |
|---|---|---|
| Random Single Agent | 38.18 ± 1.9 | 18.99 ± 1.3 |
| Single Best Agent (Oracle) | 78.32 ± 1.5 | 80.45 ± 1.1 |
| Random Delegation | 84.92 ± 0.9 | 96.46 ± 0.7 |
| REDEREF(Full) | **85.48 ± 0.8** | **96.65 ± 0.8** |

Figure 3 complements the table by visualizing mean performance with confidence intervals, reinforcing that the full REDEREFvariant consistently outperforms static and ablated baselines.

## 5.2 COLLABORATIVE GAINS

Beyond aggregate performance, we examined how collaboration among multiple agents contributes to final outcomes. As shown in Figure 4a, average score improvements are highest when 3–5 agents contribute meaningfully to the final output. This validates the intuition that synergy—not just redundancy—drives quality: a small coalition of diverse experts yields larger improvements than either a single agent or overly diffuse collaboration.

## 5.3 ABLATION: ROLE OF BELIEF UPDATES

To isolate the impact of belief-driven delegation, we compared the full system to a variant in which agents were selected uniformly at random during recursive rerouting. As shown in Table 2, disabling belief updates leads to a 17.2% increase in the use of underperforming agents and an 8.2% increase in exploratory attempts before reaching a successful outcome. Both differences are statistically significant under paired bootstrap tests ($p < 0.01$). While the number of useful contributors remains similar, the random policy wastes capacity on unreliable agents, underscoring that experience-driven learning enhances both efficiency and robustness.

Table 2: Routing behavior with and without belief updates (mean ± 95% CI).

| Method | Bad Agents | Agents Used | Agents Attempted |
|---|---|---|---|
| Random Delegation | 6.62 ± 0.4 | 4.92 ± 0.2 | 11.54 ± 0.5 |
| REDEREF(Full) | **5.65 ± 0.3** | 5.01 ± 0.2 | **10.67 ± 0.4** |

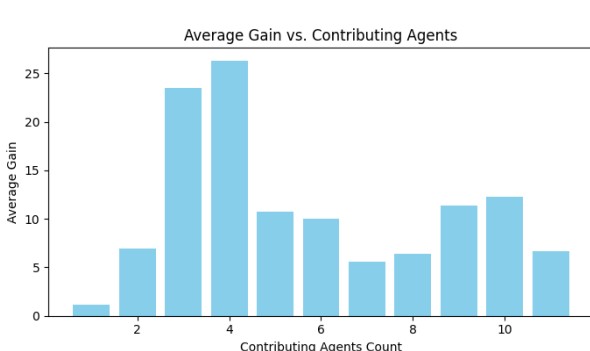 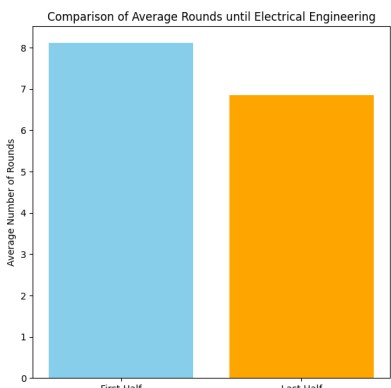

(a) Average gain vs. number of contributing agents. Collaboration among 3–5 strong contributors yields the largest quality improvements.

(b) Average rounds until Electrical Engineering expert was selected across task halves.

Figure 4: Collaboration dynamics and expert selection patterns.

## 5.4 EMERGENT SPECIALIZATION

We next examined whether agents developed stable specializations over time. In a sequence of 55 electrical-engineering tasks, the median belief score for the domain expert rose from 0.50 ($\pm 0.02$) to 0.84 ($\pm 0.03$). Concurrently, the average number of rounds required before this expert was selected declined from 8.11 ($\pm 0.4$) to 6.86 ($\pm 0.3$), as shown in Figure 4a. This trajectory demonstrates that ReDeRef not only improves aggregate performance, but also learns to preferentially route domain-specific queries to the most competent agents—evidence of emergent specialization (supporting H2).

## 5.5 ADAPTABILITY TO AGENT IMPAIRMENT

To evaluate adaptability (H3), we conducted an impairment test in which the `Biology_RAG_Agent` was replaced with systematically poor outputs after the first 50 tasks. As shown in Table 3, the agent's belief score decreased by almost 50%, and its contributions were eliminated completely in subsequent tasks. While the overall output quality decreased modestly ($84.52 \pm 0.7$ to $81.80 \pm 0.9$), the system dynamically reallocated queries to other competent agents, preventing catastrophic degradation. This rapid adjustment is also illustrated in Figure 5, which shows diverging belief trajectories for the impaired and healthy agents.

Table 3: Adaptability under impairment: Biology agent performance before and after enforced degradation (mean $\pm$ 95% CI).

| Metric | Normal | Impaired |
|---|---|---|
| Average Belief Score | $0.35 \pm 0.02$ | $0.23 \pm 0.03$ |
| Agent Contributions | $12 \pm 1.1$ | 0 |
| Final Output Score | $84.52 \pm 0.7$ | $81.80 \pm 0.9$ |

## 5.6 QUALITATIVE DYNAMICS

Qualitative inspection further illustrates ReDeRef's recursive dynamics. In one representative task, the system initially delegated to an electrical-engineering agent, which produced a technically correct but incomplete design. After a failure judgment, ReDeRef re-delegated to a narrative-oriented agent, which supplied the missing community-education perspective. The final aggregated response integrated both technical and social dimensions, and was judged successful. Such trajectories exemplify how recursive re-routing enables the system to synthesize complementary expertise and recover from early missteps.

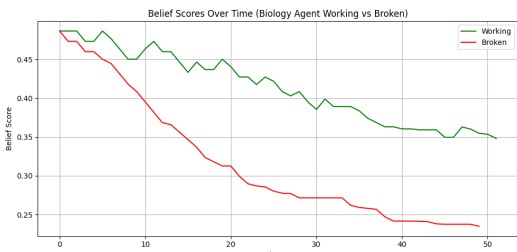

Figure 5: Belief score trajectories for Biology agent under normal vs. impaired conditions.

**Summary.** Across quantitative benchmarks and qualitative case studies, ReDeRef demonstrates clear evidence of synergy, specialization, and adaptability. The framework consistently outperforms static or naive baselines, gains quality improvements from multi-agent synergy, learns to favor domain experts over time, and dynamically reallocates away from impaired agents, providing strong empirical support for the hypotheses outlined in Section 4.

## 6  DISCUSSION AND CONCLUSION

We introduced REDEREF, a lightweight, training-free controller for multi-agent LLM systems that addresses three persistent challenges: dynamic task routing, long-horizon credit assignment, and context fragility. By combining Thompson-sampling delegation, reflection-driven updates, and memory-aware priors, REDEREF achieves higher task success and enables emergent specialization, synergy, and adaptability with minimal overhead.

**Key insights.** Our findings support a growing body of evidence that lightweight, interpretable mechanisms can rival or surpass resource-intensive pipelines. The Bayesian formulation resonates with probabilistic models of agent trust Wu et al. (2021) and bandit methods for online decision-making Chapelle & Li (2011), while avoiding the instability and sample inefficiency of deep RL Mnih et al. (2015). The recursive loop mirrors ideas from search-based reasoning such as MCTS Browne et al. (2012), yet operates at a fraction of the cost. In ablations, removing belief-guided routing increased poor-agent usage and wasted attempts, underscoring the importance of online feedback.

**Limitations.** REDEREF depends on judge reliability; biases in machine adjudication are well documented Karpinska et al. (2021) and can distort competency updates. Sequential recursion can add latency in long tasks, and early cold-start behavior resembles random delegation before memory stabilizes. These issues highlight the need for robust judges, parallelization, and improved priors.

**Broader implications.** The transparency of REDEREF is advantageous for responsible AI: Beta posteriors, verdicts, and rationales create an auditable trail Doshi-Velez & Kim (2017), contrasting with opaque RL policies or heavily finetuned controllers. However, adaptive down-weighting could prematurely exclude competent agents. Fairness audits, recalibration, and ensembles of judges are therefore important for deployment in sensitive domains such as healthcare, scientific discovery, or legal reasoning.

**Future directions.** Several extensions are promising. Hybridizing Bayesian delegation with reinforcement learning could combine fast adaptability with long-term planning Silver et al. (2017). Expanding beyond binary success/failure into richer error taxonomies would yield more informative credit assignment. Parallel delegation could mitigate latency, and programmatic verifiers or retrieval-grounded entailment models Li et al. (2023) may improve judging reliability.

**Conclusion.** Emergent and robust collaboration in LLM collectives does not require complex black-box architectures. Simple probabilistic heuristics, delegation, reflection, and memory, can transform independent agents into cohesive, adaptive systems. This "fast and frugal" approach offers a scalable, interpretable path forward for multi-agent LLM research and practice.

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

# A APPENDIX

## A.1 RESEARCH QUESTIONS AND HYPOTHESES

Our study is guided by the following research questions and the corresponding hypotheses:

**RQ1: Synergy.** Does collaboration in REDEREF produce outcomes superior to individual agents or non-adaptive baselines? **H1:** The full system will achieve significantly higher task success rates than (a) random single-agent selection, (b) the best-performing single agent, and (c) ablated system variants.

**RQ2: Specialization.** Do agents develop distinct roles through repeated interaction with task categories? **H2:** Belief scores $b_i$ will diverge over time, with consistently successful agents in a domain (e.g., mathematical reasoning) acquiring higher scores and being preferentially selected for related tasks.

**RQ3: Adaptability.** Can the routing policy adjust dynamically to shifts in agent reliability? **H3:** When a previously high-performing agent is programmatically impaired mid-experiment, its belief score and selection frequency will decline, redirecting tasks to alternative agents.

## A.2 EVALUATION TASKS

To ensure diversity and rigor, we constructed a multi-stage pipeline to generate tasks requiring multi-agent collaboration.

### A.2.1 STAGE 1: SINGLE-AGENT QUESTION GENERATION

We first generated domain-specific questions tailored to two classes of agents:

- **RAG Agents**: Retrieval-augmented models grounded in curated domain datasets (e.g., biology, finance, medicine) hug (2025).
- **Conversational Agents**: Prompt-based agents covering domains such as career guidance, fitness, or literature.

Each agent was tasked with producing 20–22 realistic questions, paired with a *Model Context Protocol* (MCP) capturing intent, tools, and plausible follow-ups. All outputs were archived in structured JSON format.

### A.2.2 STAGE 2: MULTI-AGENT TASK SYNTHESIS

Single-agent questions were combined into multi-agent tasks by sampling 4–6 diverse agents and synthesizing 15 composite tasks per batch. Each task was required to:

- Necessitate distinct, non-redundant contributions from each agent,
- Require multi-layered reasoning (planning, analysis, execution),

- Include a merged MCP integrating the intents, tools, and follow-ups of all contributing agents.

Examples include generating healthcare policy reports using medical, political, and geographic agents, or constructing recovery plans with biology, fitness, finance, and scheduling agents.

### A.2.3 STAGE 3: ITERATIVE SAMPLING

This process was repeated until the pool of questions was exhausted, yielding a benchmark suite explicitly designed to enforce distributed reasoning. Each task was unsolvable by any single agent, thereby ensuring collaborative evaluation.

### A.3 AGENT POPULATION

The agent zoo consisted of:

- **Specialist RAG Agents:** Domain-specific agents built on RAG models with access to curated knowledge bases.
- **Generalist Agents:** Base LLMs without specialized tools, responsible for generic reasoning, summarization, and synthesis.

### A.4 BASELINES AND ABLATION STUDIES

We compared REDEREF against both baseline and ablated variants.

### A.4.1 BASELINES

- **Random Single Agent:** A randomly chosen agent attempts the task once; no re-routing is permitted.
- **Single Best Agent (Oracle):** The highest-performing individual agent is identified and used in isolation. This baseline highlights the limitations of single-agent performance in multi-faceted tasks.

### A.4.2 ABLATIONS

- **Random Delegation:** The recursive loop is retained, but agent selection is uniformly random, eliminating Bayesian guidance.

### A.5 METRICS FOR EMERGENT BEHAVIORS

We adopt targeted metrics to capture performance, specialization, adaptability, and synergy.

### A.5.1 PERFORMANCE

- **Task Success Rate:** Percentage of tasks achieving a final answer score $\geq 85$ as judged by an LLM.
- **Agents Attempted:** Average number of agents invoked before reaching a solution.

### A.5.2 SPECIALIZATION (H2)

- **Belief Score Evolution:** Temporal trajectories of $b_i$ for domain experts (e.g., Math agent).
- **Rounds to Expert Selection:** Average delegation steps before the correct expert is chosen for domain-specific tasks, expected to decline as belief updates accumulate.

### A.5.3 ADAPTABILITY (H3)

- **Agent Impairment Test:** Across 100 tasks, one agent (e.g., Biology) operates normally for the first 50, then is forced to fail for the next 50.
- **Belief Score Decline:** Comparison of belief trajectories pre- and post-impairment.

- **Contribution Frequency:** Task participation counts across both phases.

- **System Output Score:** Average final answer score across both segments, reflecting resilience to impaired agents.

### A.5.4 SYNERGY (H1)

- **Synergy Gain:** Defined as

$$\text{Gain} = \text{SuccessRate(Full System)} - \text{SuccessRate(Single Best Agent)}.$$

A positive gain indicates that the system's collaborative performance exceeds that of its best individual component.

## A.6 IMPLEMENTATION DETAILS

### A.6.1 FRAMEWORK ARCHITECTURE

The REDEREF framework is implemented as a modular pipeline consisting of six core components: `WorkflowManager`, `BayesianDelegator`, `SelfReflectionStep`, `InfoMergeStep`, `MemoryUpdateStep`, and `RecursiveRouting`. The `WorkflowManager` serves as the main orchestrator, coordinating agent delegation, reflection, memory updates, and recursive re-routing according to Algorithm 1.

**Core Classes and Data Structures.** The central `Task` class maintains task state including query text, agent outputs, reflection scores, memory updates, and completion status. Agent performance is tracked through belief parameters $(\alpha_i, \beta_i)$, cooldown counters, and historical success rates. The framework supports both Bayesian delegation via Thompson sampling and traditional LLM-based ranking through a configurable `use_bayesian` parameter.

### A.6.2 BAYESIAN DELEGATION IMPLEMENTATION

Thompson sampling is implemented in the `BayesianDelegator` class with the following key features:

- **Agent Selection**: For each agent $i$, sample $\hat{\theta}_i \sim \text{Beta}(\alpha_i, \beta_i)$ and select $i^* = \arg\max_i \hat{\theta}_i$ (subject to cooldown constraints).

- **Binary Updates**: After judging, update $\alpha_{i^*} \leftarrow \alpha_{i^*} + y$ and $\beta_{i^*} \leftarrow \beta_{i^*} + (1-y)$ where $y \in \{0, 1\}$ indicates success/failure.

- **Cooldown Mechanism**: Agents are temporarily excluded for $r$ rounds after selection to encourage exploration. If all agents are cooling, the framework forces exploration by selecting the agent with the smallest remaining cooldown.

- **Belief Persistence**: Agent beliefs are stored in JSON format and loaded across sessions to maintain long-term memory.

### A.6.3 MEMORY-AWARE PRIOR INITIALIZATION

Historical performance data is used to initialize belief priors via similarity-weighted aggregation:

$$\alpha_i \leftarrow \alpha_0 + \sum_{m \in \mathcal{M}} K(\text{embed}(q), \text{embed}(q_m)) \cdot y_m \cdot \exp(-\lambda \Delta t_m) \tag{4}$$

$$\beta_i \leftarrow \beta_0 + \sum_{m \in \mathcal{M}} K(\text{embed}(q), \text{embed}(q_m)) \cdot (1 - y_m) \cdot \exp(-\lambda \Delta t_m) \tag{5}$$

where $K(\cdot, \cdot)$ is cosine similarity over sentence embeddings, $\Delta t_m$ is task recency, and $\lambda = 0.1$ controls temporal decay. This initialization reduces cold-start inefficiency by biasing selection toward agents with historically strong performance on similar tasks.

### A.6.4 MULTI-LAYERED JUDGE SYSTEM

The `SelfReflectionStep` implements a four-stage evaluation pipeline:

1. **Agent Output Scoring**: Individual agent responses are scored on a 0-100 scale using task-specific rubrics.

2. **Binary Success Evaluation**: A calibrated LLM judge determines whether the merged output satisfies task requirements, yielding $E \in \{\text{SUCCESS}, \text{FAILURE}\}$.

3. **Completeness Assessment**: The judge evaluates whether additional agent input would improve the response quality.

4. **Agent Refinement**: Underperforming agent outputs are iteratively improved based on judge critiques.

Judge calibration is performed on a held-out validation set of 200 labeled examples to estimate false positive/negative rates and set decision thresholds.

### A.6.5 AGENT ZOO SPECIFICATION

The experimental agent population consists of two classes:

- **RAG Agents**: Domain-specific agents (`ExpertAgent`) with retrieval augmentation over curated knowledge bases. Domains include mathematics, law, finance, biology, medicine, and electrical engineering. Each agent loads domain-specific datasets and uses specialized prompt templates with retrieval-grounded context.

- **Conversational Agents**: LLM-based agents (`ConversationalAgent`) without retrieval, covering fitness, literature, technology, geography, storytelling, politics, academics, career guidance, trivia, and daily planning. Agent configurations are specified in YAML format with domain-specific constraints and prompt templates.

All agents are initialized with uniform priors $\alpha_0 = \beta_0 = 1$ unless memory-aware initialization is enabled.

### A.6.6 INFORMATION MERGING AND TRUST WEIGHTING

The `InfoMergeStep` aggregates agent responses using trust-weighted selection:

1. Compute trust scores $t_i = \alpha_i / (\alpha_i + \beta_i)$ for each contributing agent.

2. Filter responses from agents marked as "bad" (belief score below threshold).

3. Merge remaining responses using LLM-based synthesis weighted by trust scores.

4. Validate merged output through evidence-grounding and consistency checks.

### A.6.7 EXPERIMENTAL INFRASTRUCTURE

The evaluation framework (`run_belief_experiment.py`) supports:

- **Configurable Agent Selection**: Systematic sampling from the agent zoo with controllable population size.

- **Question Processing Pipeline**: Batch processing with configurable delays and timeout handling.

- **Comprehensive Logging**: Results are logged to structured JSON files including initial outputs, recursive delegation traces, final merged responses, and detailed performance metrics.

- **Statistical Validation**: Built-in A/B testing, bootstrap confidence intervals, and performance benchmarking capabilities.

### A.6.8 EVALUATION METRICS IMPLEMENTATION

Quality assessment employs a multi-faceted scoring system:

- **Output Quality**: 0-100 scale scoring of initial vs. merged outputs using task-specific rubrics.

- **Quality Gains**: Absolute improvement (merged - initial) and relative improvement ((merged - initial) / initial).

- **Agent Contribution Tracking**: Classification of agents as "contributing" (positive impact) vs. "bad" (negative impact) based on comparative evaluation.

- **Routing Statistics**: Delegation depth, agent selection frequency, and belief evolution trajectories.

### A.6.9 REPRODUCIBILITY AND CONFIGURATION

All experiments are reproducible through:

- **Deterministic Sampling**: Fixed random seeds for Thompson sampling and LLM generation.

- **Configuration Management**: YAML-based agent specifications and experimental parameters.

- **Version Control**: Git-tracked experimental runs with commit hashes logged in results.

- **Environment Specification**: Docker containers with fixed dependency versions and Azure OpenAI API configurations.

### A.6.10 COMPUTATIONAL REQUIREMENTS

Typical experimental runs require:

- **Hardware**: 16GB RAM, 4-core CPU for coordination logic; GPU optional for local LLM inference.

- **API Costs**: $0.50-2.00 per task depending on recursion depth and agent complexity.

- **Runtime**: 2-5 minutes per task with Azure OpenAI; 30-60 seconds with local models.

- **Storage**: 10-50MB per 100 tasks for complete logs and belief persistence.

