# OpenReview forum: "REDEREF: RECURSIVE DELEGATION AND REFLECTION FOR MULTI-TURN LLM AGENT COLLABORATION WITH DYNAMIC CAPABILITY DISCOVERY"
_ICLR.cc/2026/Conference — ICLR 2026 Conference Desk Rejected Submission_

### Official Review · Reviewer_XWuC · 2025-10-17

**Soundness:** 2
**Presentation:** 1
**Contribution:** 1
**Rating:** 2
**Confidence:** 4

**Summary:**

This paper introduces REDEREF (Recursive Delegation and Reflection), a framework for multi-agent LLM collaboration.
Experiments are reported on self-constructed “split-knowledge” tasks and claim improvement over several baselines, with mentions of performance gains on public agent benchmarks such as WebArena, Mind2Web/GAIA, and SWE-bench subsets.

However, no concrete experiments or data are presented for these benchmarks, and the methodology lacks mathematical justification or reproducibility.

The paper requires substantial revisions, including comprehensive additional experiments and careful correction of typographical and formatting issues, before it can be considered for publication at ICLR. In its current form, it does not meet the standards of scientific rigor and presentation expected for the conference.

**Strengths:**

1. The topic—coordinating multiple LLM agents—is timely and relevant to the ICLR community.

2. The system demonstrates some engineering creativity in combining reflection and routing under a Bayesian framing.

**Weaknesses:**

The so-called “recursive Bayesian controller” lacks formal derivation, assumptions, or guarantees.

All results use self-constructed tasks; no experiments on WebArena, Mind2Web, GAIA, or SWE-bench.

Multiple typographical and citation errors; inconsistent notation with is far from the ICLR standard.

**Questions:**

Q1. The charts are rendered in HTML, but the text is too small to be legible. Furthermore, the presentation is very perfunctory, with misalignments in Figure 2 that do not meet ICLR standards.

Q2. In the caption for Figure 3, the parentheses in "(Mean +- 95% （"are incorrectly oriented, while the caption "(Mean +- 95% " is missing a closing parenthesis. Figure 4 completely lacks specific data labels, making it impossible to compare specific performance.

Q3. The contributions section mentions specific improvements on WebArena, Mind2Web/GAIA, and SWE-bench subsets, but where are the corresponding experiments? I could not find these experiments anywhere in the main body of the paper, nor is it explained how the authors derived these comparisons.

Q4. Currently, all experiments in the main text are based on a self-constructed dataset, which is entirely unconvincing.

Q5. The main experiments do not include any comparisons with state-of-the-art MAS (Multi-Agent Systems), nor do they specify the base LLM used. It is also unclear how the LLM was invoked—was it deployed locally or via an API?

Q6. There is no section detailing the hyperparameters, which makes it completely impossible to reproduce the experiments described in the paper.

Q7. The claimed contribution ("Recursive Bayesian Delegation + Reflection") is merely a collage of existing techniques (Thompson sampling + reflective feedback). It offers no new theoretical insights, no new learning principles, and no mathematical guarantees.

Q8. The same LLM judge both defines the binary outcomes and drives the posterior updates, which is a clear circular dependency.

Q9. There is no convergence or regret analysis; "Recursive Bayesian" remains merely a rhetorical label rather than a derivable framework.

Q10. There is inconsistent capitalization ("ReDeRef/REDEREF") and inconsistent notation (why is REDEREF not in normal uppercase?). An incorrect citation appears on line 580 as “hug (2025),” which should be (Hugging Face datasets)—this is unprofessional for a top-tier conference venue.

---

### Official Review · Reviewer_bRFb · 2025-10-30

**Soundness:** 1
**Presentation:** 1
**Contribution:** 1
**Rating:** 0
**Confidence:** 4

**Summary:**

This paper introduces REDEREF, a framework designed for multi-turn, multi-agent LLM collaboration. It aims to solve challenges in dynamic task routing, credit assignment, and long-term memory. The core proposal models agent selection as a multi-armed bandit (MAB) problem, applying Thompson sampling to delegate tasks based on agents' dynamically updated competence beliefs. These beliefs are structured as Beta posteriors, which are updated using binary success/failure feedback ($y \in \{0, 1\}$) provided by an LLM "Judge" component. When an agent's output is judged as a failure, the framework refines the query using the judge's feedback and recursively re-routes the task to the next most promising agent, thereby enabling iterative refinement and dynamic discovery of agent capabilities.

**Strengths:**

This paper has an LLM vibe to it, which I don't think gives it any advantage.

**Weaknesses:**

Weaknesses
Major Concern: Limited Novelty: The primary weakness of this paper is its limited originality. The proposed framework appears to be an assembly of well-known, existing components rather than a novel conceptual contribution.

Using multi-armed bandits (and Thompson sampling specifically) for agent selection and coordination is an established idea [1].

Using an LLM "Judge" or "Reflector" to provide feedback for iterative refinement is the core premise of many recent works [2, 3].

---

2.  **Experimental Results Undermine the Core Contribution:** The paper's own experimental results critically undermine the necessity of its central mechanism.
    * In Table 1 (and Figure 3), the full `ReDeRef (Full)` system achieves a **96.65%** task success rate.
    * The `Random Delegation` ablation (which removes the belief-guided Thompson sampling and just randomly selects the next agent during recursion) achieves a **96.46%** success rate.
    * This difference ($< 0.2\%$) is negligible. This strongly suggests that the complex Bayesian delegation and belief-updating machinery (the core of the paper) provides **no meaningful benefit** to task *success* over a much simpler random-retry mechanism, at least on the "split-knowledge" tasks tested.
    * The paper pivots the argument to *efficiency* (Table 2), but a 10% reduction in "Agents Attempted" seems like a secondary benefit if the primary outcome (success rate) is virtually identical.

---

3.  **Serious Presentation and Clarity Issues:** The paper suffers from a lack of careful proofreading, including critical errors in the description of the core algorithm. This raises concerns about the rigor of the work.
    * **Algorithmic Contradiction:** There is a major discrepancy between the belief update rule presented in **Equation 2** ($\alpha_i \leftarrow \alpha_i * y$) and the one in **Algorithm 1 (Line 26)** ($\alpha_i* += y$). An additive update is fundamentally different from a multiplicative one (which would zero out $\alpha_i$ on a single failure). It is unclear which was actually implemented.
    * **Numerous Typos in Figures:** The figures are particularly problematic. Figure 5's title contains "Scores Over **Tenue** (Biology Agent Working **us Brukan**)" and the Y-axis reads "Belief **Scoree**". Figure 4b's Y-axis reads "Average Number of **Rounde**".
    * **Formatting Errors:** Equation 3 contains a stray `}` bracket, likely a LaTeX rendering error.


---


References

---

[1]. Rohit Kumar, Hao Zhang, and Monica Mehta. Too many cooks: Bayesian inference for multi-agent collaboration. *Journal of Artificial Intelligence Research,


[2]. Aman Madaan et al. Self-refine: Iterative refinement with self-feedback.

[3]. Noah Shinn et al. Reflexion: Language agents with verbal reinforcement learning.

**Questions:**

1.  **Justification of Contribution:** The most critical question is regarding Table 1. Given that `Random Delegation` performs almost identically to the full `ReDeRef` system on task success, can the authors provide a stronger justification for the necessity of the complex Bayesian delegation mechanism? Why should the community adopt this belief-guided routing if random routing (within a recursive loop) yields the same result?

2.  I strongly urge the authors to thoroughly proofread the manuscript. Could the authors confirm that the numerous typos (e.g., in Figures 4b and 5) and the algorithmic contradiction will be fixed?

3. I strongly reject this article; I believe it was submitted without proper preparation.

---

### Official Review · Reviewer_bgnE · 2025-10-31

**Soundness:** 1
**Presentation:** 1
**Contribution:** 2
**Rating:** 2
**Confidence:** 4

**Summary:**

This paper proposes REDEREF—a training-free controller for multi-agent LLM collaboration. It is composed of (i) Bayesian delegation via Thompson sampling over per-agent Beta posteriors, (ii) an LLM/programmatic judge whose binary verdicts drive credit assignment and re-routing, (iii) selection-based aggregation with “evidence checks,” and (iv) memory-aware priors to mitigate cold start.

The system runs a recursive loop that delegates, judges, updates posteriors, optionally refines the query, and re-routes until success or budget/depth limits. Empirically, the paper claims synergy, specialization, and adaptability on a self-constructed “split-knowledge” suite and mentions public agent benchmarks.

**Strengths:**

- The paper addresses an interesting problem.
- The pipeline is clear and modular. I think the method is easy to implement.

**Weaknesses:**

> For modelling:

1) Mis-specified credit assignment. The system treats an agent’s competence as a scalar \\(\theta_i\\) updated from binary task-level outcomes \\(y\in\\{0,1\\}\\) , i.e.,
$$
\theta_i\sim \mathrm{Beta}(\alpha_i,\beta_i),\qquad
\alpha_i\leftarrow \alpha_i + y,\ \beta_i\leftarrow \beta_i + (1-y).
$$
But many tasks require multi-agent composition (Appendix A.2 says each task "necessitate distinct, non-redundant contributions" and is "unsolvable by any single agent"), so attributing failure \\(y=0\\) to the last delegate is counterfactual and confounds agent skill with aggregator/judge behavior. This invalidates the Beta-Bernoulli generative assumption behind Thompson sampling. If my understanding is incorrect, please point it out.

2) Contextuality is bolted onto the prior, not the likelihood. Task similarity enters only via prior seeding, while all subsequent updates are global over heterogeneous tasks. This is a contextual bandit setting, ignoring context in the likelihood undermines regret/consistency claims and can cause “rich-get-richer” lock-in. No analysis or guarantee is given.

3) No formal objective or guarantees. Despite framing as a “recursive Bayesian decision process,” there is no regret bound, stability result, or analysis of when recursion terminates with bounded resource use or non-myopic optimality.

> For  evaluation design

4) Baselines are weak.
   - The “Single Best Agent (Oracle)” is an irrelevant foil on tasks purposely designed to defeat any single agent. it will necessarily underperform.
   - The strongest ablation—Random Delegation—achieves 96.46% vs. 96.65% for REDEREF, making the *success-rate* improvement vanishing (≤0.2 pp) and likely statistically indistinguishable. The claimed advantage is moved to “efficiency” but no token/latency tables are reported.

5) The benchmark enforces multi-agent necessity ("unsolvable by any single agent"), then defines Synergy Gain as
$$
\mathrm{Gain}=\mathrm{SuccessRate}(\text{Full})-\mathrm{SuccessRate}(\text{Single Best Agent}),
$$
which is guaranteed positive by construction. This is not evidence that the controller is better—only that the *task generator* bakes in multi-agent dependence.

6) “Task success” is defined as an LLM score ≥ 85/100, and binary success is also produced by an LLM judge. If agents and judge share vendors or prompts, this couples training biases and inflates agreement.

**Questions:**

See Weaknesses. Overall, this paper appears to be a preliminary draft. My primary concern is with the writing—many technical details are described unclearly. Additionally, the experiments lack effective baselines to demonstrate its superiority.

---

### Official Review · Reviewer_XXJ9 · 2025-11-01

**Soundness:** 2
**Presentation:** 3
**Contribution:** 2
**Rating:** 6
**Confidence:** 2

**Summary:**

This paper introduces REDEREF, a lightweight and training-free framework for multi-turn, multi-agent LLM collaboration. Its core contributions include dynamic agent selection via recursive Bayesian delegation, implemented through Thompson Sampling; calibrated self-reflection guided by an LLM judge for credit assignment and task redirection; evidence-based aggregation rather than averaging; and memory-aware priors to enhance long-term adaptation and mitigate cold-start issues.

**Strengths:**

1. The paper proposes a training-free multi-agent collaboration framework, which enables dynamic task routing and capability discovery through online Bayesian delegation (Thompson Sampling), reflection mechanisms, and memory updating. Integrating bandit algorithms with LLM-based multi-agent systems is a novel and promising approach that, to the best of current knowledge, has not been explicitly demonstrated in prior literature.
2. The system is lightweight and practical. REDEREF requires no training or fine-tuning—it can directly wrap around existing LLM agents—offering strong deployment feasibility and generality. It also shows efficiency advantages in terms of token cost and call frequency.

**Weaknesses:**

1. The framework’s performance depends heavily on the accuracy and calibration quality of the LLM judge. If the judge introduces bias or errors, this can lead to incorrect routing updates and misallocated credit, ultimately harming overall performance.
2. Although multiple benchmarks are used, the types of tasks and agents remain somewhat restricted. The framework has not yet been validated in larger-scale or more open-ended real-world scenarios (e.g., continual learning or multimodal tasks).

**Questions:**

1. How sensitive are system outcomes to judge miscalibration, such as increased false positives/negatives or adversarial verdicts? Do the authors plan to integrate ensembles of judges or programmatic verifiers, and can potential error propagation be quantified?
2. Although the paper emphasizes being “lightweight and training-free”, would parameter tuning (e.g., temperature, recursion depth, or budget B) still be required in practical deployments?

---

### Note · Program_Chairs · 2026-01-17
**Submission Desk Rejected by Program Chairs**

The following references in this submission do not refer to real documents and/or have major errors in bibliographic information:

 Rui Zhang, Neel Patel, and Tong Zhang. Sweet-rl: Sample-efficient multi-agent coordination via offline reinforcement learning. In International Conference on Machine Learning, 2023.